# Novel Nickel(II), Palladium(II), and Platinum(II) Complexes with *O*,*S* Bidendate Cinnamic Acid Ester Derivatives: An In Vitro Cytotoxic Comparison to Ruthenium(II) and Osmium(II) Analogues

**DOI:** 10.3390/ijms23126669

**Published:** 2022-06-15

**Authors:** Jana Hildebrandt, Norman Häfner, Helmar Görls, Marie-Christin Barth, Matthias Dürst, Ingo B. Runnebaum, Wolfgang Weigand

**Affiliations:** 1Institut für Anorganische und Analytische Chemie Friedrich-Schiller Universität Jena, Humboldtstraße 8, 07743 Jena, Germany; jana.hildebrandt@astrazeneca.com (J.H.); helmar.goerls@uni-jena.de (H.G.); marie-christin.barth@uni-jena.de (M.-C.B.); 2Department of Gynecology, Jena University Hospital—Friedrich-Schiller University Jena, Am Klinikum 1, 07747 Jena, Germany; norman.haefner@med.uni-jena.de (N.H.); matthias.duerst@med.uni-jena.de (M.D.)

**Keywords:** metal-based compounds, cancer treatment, platinum resistance, ovarian cancer

## Abstract

(1) Background: Since the discovery of cisplatin’s cytotoxic properties, platinum(II) compounds have attracted much interest in the field of anticancer drug development. Over the last few years, classical structure–activity relationships (SAR) have been broken by some promising new compounds based on platinum or other metals. We focus on the synthesis and characterization of 17 different complexes with β-hydroxydithiocinnamic acid esters as *O*,*S* bidendate ligands for nickel(II), palladium(II), and platinum(II) complexes. (2) Methods: The bidendate compounds were synthesized and characterized using classical methods including NMR spectroscopy, MS spectrometry, elemental analysis, and X-ray crystallography, and their cytotoxic potential was assessed using in vitro cell culture assays. Data were compared with other recently reported platinum(II), ruthenium(II), and osmium(II) complexes based on the same main ligand system. (3) Results: SAR analyses regarding the metal ion (M), and the alkyl-chain position (P) and length (L), revealed the following order of the effect strength for in vitro activity: M > P > L. The highest activities have Pd complexes and ortho-substituted compounds. Specific palladium(II) complexes show lower IC_50_ values compared to cisplatin, are able to elude cisplatin resistance mechanisms, and show a higher cancer cell specificity. (4) Conclusion: A promising new palladium(II) candidate (Pd3) should be evaluated in further studies using in vivo model systems, and the identified SARs may help to target platinum-resistant tumors.

## 1. Introduction

Cisplatin was first synthesized by M. Peyrone in 1845, and its anticancer properties were discovered accidentally by B. Rosenberg and coworkers in 1965 [1]. Rosenberg´s discovery led to the approval of the drug by the FDA in 1979 [2]. The proposed mechanism of action involves binding to its main target (DNA) through the specific DNA-base guanine, and the formation of intra- and inter-strand adducts. Adducts lead to distortions of the helical DNA structure, DNA damage, the disturbance of DNA replication and transcription, and the activation of several intracellular signal pathways potentially inducing apoptosis [3,4,5,6,7]. Cisplatin-based therapy is limited by its toxic side effects, the low selectivity of the drug, and resistance mechanisms [2,5,8,9]. Therefore, soon after cisplatin’s development, the design of new platinum(II) drugs began, leading to the worldwide approval of carboplatin and oxaliplatin [2,5]. Both drugs follow the structure–activity relationships (SARs) of cisplatin: they are square-planar neutral cis-platinum(II) complexes with inert ammine or chelating diamine ligands, and two semi-labile chlorides or oxygen-coordinated bidendate ligands. Thus, they are likely affected by the same resistance mechanisms and cause similar side effects, although oxaliplatin may evade some resistance mechanisms [2,8,10]. Specifically, oxaliplatin adducts are more bulky, differentially recognized by DNA repair systems, and not affected by resistance caused by mismatch repair deficiency [10,11]. However, a systematic review identified a poor response rate to oxaliplatin in cisplatin-resistant or -refractory cancers in accordance with most preclinical studies, although some models that are highly resistant to cisplatin show a response to oxaliplatin [12]. 

This led to a rethinking of the traditional platinum SARs, and to several new compounds which do not follow these rules and are designed for improved cytotoxic activities, using platinum(II) or platinum(IV) as the core of the drug [8,13,14,15,16]. An excellent review written by Lippard and coworkers classifies new platinum(II) and platinum(IV) molecules in three classes: classical, non-classical, and nanodelivery molecules [14]. Classical platinum(II) molecules are normally designed to follow the SARs of cisplatin and its analogues, but also target specific structures on cancer cell surfaces, therefore enhancing the cellular uptake and the selectivity of the drugs [14]. Non-classical compounds are designed to focus on different mechanisms of action, e.g., trans-compounds or monofunctional complexes, as well as platinum(II) molecules which do not bind covalently to the DNA, for example metallointercalators, which are able to intercalate in the DNA [2,14,17]. We recently reported on platinum(II) complexes bearing an *O*,*S*-bidendate ligand, DMSO (dimethylsulfoxide) and one chloride as a leaving group with promising results on cisplatin resistant cell lines, and on their interaction with targets other than the DNA [18,19,20].

In addition to improvements in platinum drugs, intensive research has aimed to develop new complexes using other metal ions (copper, nickel, gold, ruthenium, osmium, and palladium). Several recent reviews focus on these attempts at new metal-based drug design, e.g., ruthenium-, osmium-, and palladium-based molecules [21,22,23,24,25,26,27,28]. 

Palladium complexes are among the most widely investigated molecules for anticancer drug design [21]. Palladium(II) is a d^8^-system similar to platinum(II), and therefore structural analogues of platinum(II) complexes have been synthesized and their anticancer properties explored [21,29,30]. However, those analogues do not show activity comparable to their promising platinum(II) counterparts, as the ligand exchange kinetics for palladium(II) complexes are 10^5^ times faster. Therefore, these compounds hydrolyze quickly after injection, interact with several biomolecules, leading to inactivation, and do not reach their final target [13,21,31,32,33]. Thus, the palladium analogues of cisplatin and carboplatin show no antitumor activity [13]. Changing the traditional cisplatin-based SARs is crucial for the design of potent palladium(II) complexes. In 2016, Huq and coworkers reported a SARs guide for palladium(II) complexes while comparing published data for 847 palladium complexes [21]. They concluded that palladium(II) compounds follow different rules than platinum(II) complexes, and the most promising candidates often show specific structural characteristics. Enhanced antitumor activity was detected with bulky, chelating ligands and a high lipophilicity. Moreover, ortho-substituted benzyl-rings on ligands exhibited better cytotoxic properties than their meta-/ para-substituted analogues [21]. A high activity of Pd compounds can result from increased DNA binding activity, but can also be related to different modes of action that may involve protein binding, endoplasmatic stress induction, or mitochondrial targeting [34,35,36,37,38].

Nickel is another metal that can form complexes with organic ligands, but is not as well studied as Pt, Pd, Ru, or Cu compounds for cytotoxic activity. Studies focusing on the comparison of nickel(II) complexes and their platinum, palladium, and copper analogues show acceptable but no outstanding cytotoxic activity for nickel complexes [39,40]. Nickel may still have some pharmacological properties which are useful for anticancer drug design because many classes of metalloproteins exhibit nickel-ions [40]. Additionally, Ni compounds may exhibit both DNA-damaging activity and also a DNA-independent mode of action, e.g., reactive oxygen species (ROS) induction [41,42]. 

Applying the well-accepted approach of designing potential metal-based anticancer drugs with SARs other than cisplatin, we report on new platinum(II), palladium(II), and nickel(II) complexes with β-hydroxydithiocinnamic acid esters as bidendate *O*,*S*-chelating ligands. Our aim was to determine and compare the activity of these non-classical complexes and to identify the most suitable β-hydroxydithiocinnamic acid ester as ligand. We previously reported on the synthesis of this group of compounds in general [43,44,45,46,47,48], and, in the present work, add new insights into their cytotoxic activity, as well as their characteristics, including molecular structures and stability determinations. Recently, a novel mixed platinum(II) complex with an *O*,*S*-chelating ligand and the general formula [Pt(PPh_3_)_2_(L-*O*,*S*)]PF_6_ (L-*O*,*S* = *N*,*N*-morpholine-*N*′-benzoylthiourea) has been synthetized and tested, and proves to be active against tumor cells. The choice of the β-hydroxydithiocinnamic acid ester as the ligand system is based on our described promising results for ruthenium(II) and osmium(II) complexes also bearing this *O*,*S*-bidendate ligand [49,50]. 

Figure 1 shows an overview of the compounds discussed in this work. Both platinum-sensitive and -resistant epithelial ovarian cancer (EOC) cell lines were chosen as models for the in vitro comparison of the compounds’ cytotoxic effects. EOC is a leading cause of death in women with gynecologic cancer (approx. 220,000 new cases annually worldwide) [51]. Standard care comprises cytoreductive surgery, combined with chemotherapy using a platinum-based regimen in combination with other cytotoxic drugs, plus molecularly targeted strategies for maintenance therapy. While EOC is, in the majority of cases, a platinum-sensitive disease, eventually the majority of patients will relapse and develop a platinum resistance. Platinum resistance is the main challenge to a long-lasting successful therapeutic effect, thus contributing to the low five-year survival rate of approximately 40% [51].

## 2. Results and Discussion

### 2.1. Synthesis and Characterisation

All β-hydroxydithiocinnamic acid esters L1–L6 were synthesized and characterized as described previously [18,50]. Ligands L1–L6 were diluted in 15 mL acetonitrile in a flask and deprotonated with sodium acetate, and the corresponding metal salt was added. The reaction mixture was stirred for 15 h at room temperature, followed by filtration and washing steps with pentane and acetonitrile (Figure 1).

Generally, characterization with ^1^H and ^13^C{^1^H} NMR spectroscopy shows results comparable to those published previously for similar Ni, Pd, and Pt complexes [46]. Table 1 displays compound M2 as an example, showing four chosen signals in the same range for the three metal complexes (M = Ni, Pd, Pt). Compared to L2, a high-field shift is observable for ^13^C signal 2, due to the complexation of the metal via the thiocarbonyl carbon, and the resulting shield of the carbon atom. Complexation results in a low-field shift for ^13^C signal 4, as the oxygen atom exhibits a σ-donor character. Interesting changes are observable for the methine proton, signal 1. Whereas Ni2, Pd2, Pt2, and PtDMSO2 show a shift to higher ppm values compared to L2, the opposite is shown for Ru2 and Os2 [18,50]. This is potentially caused by the better donor ability of the cymene ligand. The chemical structures of Ru2, Os2, and Pt2 are shown in the Appendix A. Compared to PtDMSO2, a platinum(II) complex with one L2 as a bidendate ligand, DMSO, and a labile chloride ligand, the ^1^H methine signals for Ni2, Pd2, and Pt2 are shifted around 0.2 ppm up to higher field [18]. 

The mass spectra for all nickel, palladium, and platinum complexes show molecular ions including the unique isotope pattern for Ni, Pd, or Pt, as well as fragments originating from α-cleavages specific to the β-hydroxydithiocinnamic acid esters, as described previously [18].

With the help of ^1^H NMR spectroscopy, the stability of the complexes was studied. We did not detect any degradation for Ni and Pd complexes, and only minor degradation for Pt complexes. Experiments were carried out at room temperature using DMSO-d_6_ or dichloromethane as solvents, and at 37 °C in DMSO-d_6_, showing the same results. Examples (37 °C, DMSO-d_6_, 48 h measurements) are shown in the Appendix A. 

In addition, we carried out stability measurements for Ni3, Pd3, and Pt3 (100µM) using UV–VIS spectroscopy in different buffers at room temperature (Appendix A). Measurements in 100% DMSO confirmed the stability determined by NMR. In the analyzed buffers (10% DMSO with: PBS, 120 mM NaCl, 12 mM NaCl), the compounds precipitated within the analyzed time span (11 h), resulting in a general decrease in absorbance. However, we did not observe strong changes in the spectra that would point to a decomposition of the compounds. In both NaCl buffers there was a slight absorbance increase at higher wavelengths, and this effect was stronger in Pd3 than in Pt3. Interestingly, the compounds did not precipitate in 10% DMSO supplemented with bovine serum albumin (BSA), potentially due to protein binding. This effect could prevent precipitation in the cell culture medium (supplemented with fetal calf serum), and may result in a steady release of the compounds over time.

### 2.2. Molecular Structures

Nickel(II) complexes Ni1, Ni3, Ni4, and Ni6, as well as palladium(II) complex Pd1, were characterized by means of single crystal X-ray structure determination. Figure 2 and Table 2 show the molecular structures and characteristics of Ni1 and Pd1. Data for the other nickel(II) complexes and further bond lengths and angles, as well as data for PtDMSO8, are shown in the Appendix A.

The bond lengths and angles of the nickel(II) and palladium(II) complexes are in good agreement with previously reported values [18,46]. As the structures are quite symmetric, all bond lengths and angles are in the same range for both β-hydroxydithiocinnamic acid esters cis-coordinated around the square-planar metal(II) center. Therefore, only one value is chosen for each discussion. The bond lengths of the heteroatoms (O and S) to the metal increase in the order Ni1 {O(1)-M(1): 1.8466(17), S(1)-M(1): 2.1429(7)} < Pd1 {O(1)-M(1): 2.023(4), S(1)-M(1): 2.2307(16)} < PtDMSO1 {O(1)-M(1): 2.015(7), S(1)-M(1): 2.251(6) Å} (Table 2). The O(1)-M(1)-S(1) and O(3)-M(1)-S(3) (M = Ni, Pd) angles are around 90°, resulting in a slightly distorted square-planar environment for the nickel(II) and palladium(II) compounds. Bond angles O(1)-M(1)-S(3) and S(1)-M(1)-O(3) (M = Ni, Pd) show angles of almost 180°, which is characteristic of the square-planar coordination sphere [46].

The molecular structure of L1, as well as changes in bond lengths and angles after coordination to its corresponding complex PtDMSO1, has been discussed previously [18]. Changes in bond lengths and angles for the bischelate complexes Ni1 and Pd1 are comparable to the changes for those in PtDMSO1 (Table 3). Coordination of L1 to M1 (M= Ni, Pd, Pt) results in an elongation for the C(1)-S(1)-bonds, increasing in the order of Ni1 < Pd1 < PtDMSO1. This tendency has already been observed in a high-field shift for the -C=S-group in the ^13^C{^1^H} NMR spectra. For the C(3)-O(1)-bond, a shortening can be observed, which follows the reverse order of the elongation described before, as can the described low-field shift in the corresponding ^13^C{^1^H} NMR spectra.

### 2.3. Biological Behavior

A further aim of this study was to characterize all metal complexes for their cytotoxic properties in vitro, and to determine structure–activity relationships. Therefore, all compounds were tested against a panel of cancer cell lines with different sensitivity to cisplatin: the ovarian cancer cell lines SKOV3/SKOV3cis and A2780/A2780cis [52,53], and lung cancer cell line A549. Selected compounds were additionally tested against non-cancerous cells: keratinocytes, fibroblasts, and MCF10A. Due to the low solubility of the new compounds in water, DMSO was used as a solvent. The toxic influence of DMSO was determined earlier, and experiments were carried out with 0.5 % DMSO in cell culture media and used as a reference in MTT assays (see Section 3) [18]. The conditions of these experiments were the same as for PtDMSO, Ru(II), and Os(II), and have been published [18,49,50]. Thus, alongside comparisons of different metals and different substitution patterns of the ligands described in this paper, comparisons to the other already published systems are possible. Additionally, the IC_50_ values of the different β-hydroxydithiocinnamic acid ester ligands have previously been evaluated [50].

All IC_50_ values for the 17 metal(II) complexes, as well as the reference cisplatin (CDDP), are displayed in Table 4, and an exemplary dose–response curve is shown in Appendix A. Resistance factors (RF = IC_50_^resistant cells^ / IC_50_^sensitive cells^) for the ovarian cancer cell lines have been determined. Cisplatin IC_50_ values for resistant cell lines (SKOV3cis and A2780cis) are increased compared to the sensitive cell lines (3.8 μM vs. 13.5 μM and 1.3 μM vs. 6.1 μM, respectively), resulting in high resistance factors (3.6 and 4.7, respectively). For novel metal(II) complexes, resistance factors lower than 1 show that the compounds’ activity is not affected by the cisplatin-induced resistance. For cell line SKOV3 this is proved for: Ni2, Ni4, Ni6, Pd1, Pd2, Pd4, Pd5, Pd6, Pt3, and Pt4 (total: 10 of 17 compounds). The same can be observed in A2780 for: Ni3, Ni5, Pd6, Pt2, and Pt4 (total: 5 of 17 compounds). Moreover, it is shown that most resistance factors are much lower than those of cisplatin. Taking into account only the ability to elude cisplatin resistance, Pd6 and Pt4 display the best results (Table 4) as they have lower resistance factors on both cell line pairs. The comparison of the IC_50_ values for each cell line shows the highest activity for compound Pd3, which is more active than cisplatin on four of the five cell lines, resulting in the lowest mean IC_50_ value (Figure 3A). In Table 4, all IC_50_ values lower than IC_50_ of the reference cisplatin are marked in red, highlighting that palladium complexes specifically exhibit high activity. In addition to the single IC_50_ values (Table 4), we calculated compound-specific or metal-specific mean values for all or specific groups of cell lines (Figure 3 and Figure 4). The high variance of mean values caused by the heterogeneity of cell-line-specific sensitivity prevents significant differences. However, this may reflect the clinical situation if no predictive biomarker is available, and comparisons of metal-specific mean values provide information about the general effects of different metal ions. Figure 3 depicts all compounds, ordered by increasing mean IC_50_ values. It can be concluded that Pd3, Pd4, and Pd1 are the most active compounds included in this study, followed by Ni1 and Pt4. Pd compounds exhibit a mean IC_50_ value (all six compounds, all five cell lines) that is lower compared to the mean IC_50_ for cisplatin (all five cell lines) pointing to a generally high cytotoxic activity of these complexes (Figure 4A). Moreover, it is shown in Figure 4B that the mean IC_50_ value on both resistant cell lines for Ni and Pd complexes are lower than that of cisplatin. These compounds (i.e., Pd) act specifically on the resistant cell lines, and may be an alternative option for cisplatin resistant tumors in anticancer therapy. This points to another mode of action for both Pd and Ni compounds (for discussion, see below). Both the high cytotoxic activity of Pd compounds against cisplatin-resistant cell lines and their non-superior activity against cisplatin-sensitive cell lines have previously been described [54,55,56,57,58]. Moreover, a multinuclear Pd(II) complex can potentially improve the treatment of cancer stem cells that are more resistant to platinum [59].

As mentioned above (see Introduction), Huq and coworkers reported some general structure–activity relationships (SARs) for palladium(II) complexes [21] in 2016. They proposed a higher activity for ortho-substituted phenyl rings. The top five compounds (regarding the mean IC_50_ values) of this work are: Pd3 (-*p*-OMe), Pd4 (-*o*-OEt), Pd2 (-*m*-OMe), Ni1 (-*o*-OMe), and Pt4 (-*o*-OEt). This is in good agreement with the hypothesis, as it shows three ortho-substituted metal(II)-complexes in the five most active compounds. Evaluating the top 10 compounds of this work (number 6–10: Pd1 (-*o*-OMe), Ni3 (-*p*-OMe), Ni4 (-*o*-OEt), Pd6 (-*p*-OEt), and Pt1 (-*o*-OMe) results in six ortho-substituted, three para-substituted and one meta-substituted compounds, favoring ortho-substituted ligands as most active. Moreover, calculating the percentage of activity relative to the mean of specific substance groups (e.g., relative percentage of activity of Ni compounds with ortho-substituted methoxy, or ethoxy ligand relative to the mean IC_50_ of all Ni compounds with methoxy or ethoxy ligands, respectively) proves the higher activity of ortho-substituted complexes (Figure 3B). Metal complexes with ortho-substitution are significantly more active than para- or meta-substituted ones. This difference in activity must be related to the behavior of the complexes because the ligands themselves have similar activities (Figure 3B). Thus, the presented data support the results regarding SARs for Pd complexes from Huq et al. [21] and point to similar relationships for other metal compounds (e.g., Ni, Pt). Regarding the length of the chain (methoxy vs. ethoxy group), there is no clear correlation seen for the metal complexes in general. However, Pt complexes with longer alkyl chains (ethoxy group) are significantly more active than the complexes with a methoxy residual (Figure 3C). The same effect is seen for the β-hydroxydithiocinnamic acid esters where compounds 4–6 (ethoxy group) are significantly more active than 1–3 (methoxy group; Figure 3C and Figure 4C). Nevertheless, the activity of specific compounds is affected by the combination of all characteristics. The effect strength seems to be metal ion > substitution position > alkyl-chain length (Figure 3), and the most promising candidate compared to cisplatin is Pd3, which bears a para-methoxy group at the phenyl ring.

The most active compounds for each metal, Pd3, Ni1, and Pt4, have been tested against non-cancerous cells’ keratinocytes, fibroblasts, and MCF10a to evaluate their toxicity in general (Table 5). It is known that cisplatin shows toxic side effects by interacting with normal proliferating cells. This is proven by our experiments, which show low IC_50_ values for CDDP on these cells. Specifically, the most active metal(II) complexes included in this work do not attack those cells, and it can be concluded that these complexes may show a higher selectivity for cancer cells. The importance of both the detected high activity (against cisplatin resistant cells) and the high selectivity are potentially affected by the limitations of this study. First, these data have to be validated in vivo to make conclusions about clinical benefit and use. Secondly, the unknown stability in biological systems/buffers limits our knowledge about the active species. Although we measured a high stability of the metal complexes in DMSO-d6 over 48h by NMR spectroscopy (Appendix A), earlier data of Pt-complexes with the *O*,*S*-bidendate ligand, DMSO, and chloride showed a reduced stability in biological buffer solutions [18]. Thus, we cannot exclude speciation processes and a certain contribution of specific degradation/speciation products to the biologic activity. Nevertheless, this may not affect the main results.

For compound 2 (meta-OMe), a comparison of the β-hydroxydithiocinnamic acid ester (L2), the nickel(II), palladium(II), and platinum(II) complexes of this work (Ni2, Pd2, Pt2), the previously reported platinum(II) complex with one *O*,*S*-bidendate ligand, DMSO, and chloride as additional ligands (Ptdmso2), and the corresponding ruthenium(II) and osmium(II) complexes, could be conducted (structures of PtDMSO2, Ru2, and Os2 areshown in Appendix A) [18,50]. Table 6 and Appendix A show the IC_50_ values for all 2 compounds, as well as the reference cisplatin on the five cell lines. All metal(II) compounds show lower IC_50_ values than the free β-hydroxydithiocinnamic acid ester L2. In general, the ligands themselves exhibit a lower activity than the complexes (Figure 4C). Compounds Os2 and Pd2 show the best results and lower IC_50_ values than cisplatin. For the platinum(II) complex, it can be concluded that it exhibits a lower activity not superior to the reference. However, the data show that the resistance factors for all substances are lower than for cisplatin, proving that the β-hydroxydithiocinnamic acid esters and their metal complexes are able to elude the cisplatin resistance mechanisms of ovarian cancer cell lines in vitro. Moreover, Ni2, Pd2, and Os2 are even more active compared to cisplatin in SKOV3cis, whereas the two most active compounds in A2780cis are Os2 and Pd2, showing lower IC_50_ values than the reference substance. Thus, the presented data show that complexes of metals with β-hydroxydithiocinnamic acid ester are not affected by the resistance mechanisms of cisplatin and the compounds likely have another mode of action. Ru(II) complexes exert their activity by impacting the DNA itself, and the mitochondrial activity, autophagy pathway, ROS generation, and ROS mediated apoptosis [23,24,60]. Our analyses point to a DNA-independent induction of cell death, potentially mediated by protein interactions [49,50]. Similarly, the mode of action of Os, Pd, and Ni compounds is described as both DNA-directed and DNA-independent (e.g., ER-stress induction, protein targeting, ROS generation) [28,35,37,38,39,41,42]. In addition, Ru(II) and Os(II) compound antitumor activity and specificity can be increased by redox modulators [60,61]. However, both Ru(II) and Os(II) compounds can also act independent of ROS by inhibiting protein synthesis [62,63]. Specifically, it can be suggested that the DNA-independent mechanisms are responsible for the high activity against cisplatin resistant cells. These modes of action should be analyzed in detail to improve the treatment of cancer patients. Moreover, metal-based nanoparticles, photoactivated chemotherapy, catalytic active compounds, or complexes with bioactive ligands may lead to new therapeutics and improved outcomes for cancer patients [64,65,66,67,68]. 

## 3. Materials and Methods

### 3.1. Materials and Techniques

For NMR spectroscopy, a Bruker Avance 200 MHz, 400 MHz, or 600 MHz spectrometer was used. Chemical shifts referring to SiMe_4_ are stated in ppm. Mass spectra were measured with a Finnigan SSQ 710 single quadrupole mass spectrometer operating with the direct electron ionoization at 70 eV. Elemental composition was detected with a Leco CHNS-932 apparatus. For column chromatography, Silica gel 60 (0.015–0.040 mm) was used, and TLC was performed using Merck TLC aluminum sheets (Silica gel 60 F_254_). Chemicals were ordered from Aldrich, Acros, or Fisher Scientific, and were used without additional purification. Prior to use, all solvents were dried and distilled according to standard procedures.

### 3.2. Synthesis

Different β-hydroxydithiocinnamic acid alkyl esters were prepared as described before [18,50]. The (PhCN)_2_PtCl_2_/ (PhCN)_2_PdCl_2_ as starting materials were prepared using modified literature methods [69,70].

General Procedure 1: Metal(II) Complexes with β-Hydroxydithiocinnamic Acid Alkyl esters as Ligands (M1–M6)

The corresponding ligand L1–L6 (2 equiv.), corresponding metal salt (1 equiv.), and sodium acetate (2 equiv.) were dissolved in 15 mL acetonitrile and stirred for 15 h at rt. The precipitated red crystals were filtered, washed with acetonitrile and pentane, and dried under reduced pressure.

[Ni(1-(2-methoxyphenyl)-3-(methylthio)-3-thioxo-prop-1-en-1-olate-*O*,*S*)] (Ni1)

Synthesis was performed according to general procedure 1. NiCl_2_·6H_2_O (250 mg, 1 mmol) and L1 (506 mg, 2 mmol) were used. Yield: 240 mg (44.8 %) as red crystals. ^1^H NMR (400 MHz, CDCl_3_): δ = 2.59 (s, 6H, -SC*H*_3_); 3.88 (s, 6H, -OC*H*_3_); 6.89–6.97 (m, 4H, -Ar-*m*-H); 7.28 (s, 2H, =C*H*); 7.40 (m, 2H, -Ar-*p*-H); 7.34 (d, ^3^*J_H-H_*=7.4 Hz, 2H, -Ar-*o*-H). ^13^C{^1^H} NMR (101 MHz, CDCl_3_): δ = 16.9 (-S*C*H_3_); 55.8 (-O*C*H_3_); 111.7 (-Ar-*m*-C); 115.1 (=*C*H); 120.8 (-Ar-*m*-C); 127.9 (-Ar-C1); 131.4 (-Ar-*m*-C); 132.2 (-Ar-*o*-C); 157.2 (-Ar-*C*-OCH_3_); 178.1 (-*C*-O-); 181.4 (-*C*=S). Electron ionization mass spectrometry (MS (EI)): *m*/*z* = 536 [M(^58^Ni)]^•+^. Elemental analysis: calculated for C_22_H_22_O_4_NiS_4_ C: 49.18 %; H: 4.13 %, found: C: 49.36 %; H: 4.14 %.

[Ni(1-(3-methoxyphenyl)-3-(methylthio)-3-thioxo-prop-1-en-1-olate-*O*,*S*)] (Ni2)

Synthesis was performed according to general procedure 1. NiCl_2_·6H_2_O (250 mg, 1 mmol) and L2 (506 mg, 2 mmol) were used. Yield: 300 mg (56.0 %) as red crystals. ^1^H NMR (400 MHz, CDCl_3_): δ = 2.65 (s, 6H, -SC*H*_3_); 3.86 (s, 6H, -OC*H*_3_); 7.10 (d, ^3^*J_H-H_*=8.8 Hz, 2H, -Ar-*o*-H); 7.16 (s, 2H, =C*H*); 7.40 (m, 2H, -Ar-*p*-H); 7.34 (m, 2H, -Ar-*m*-H); 7.47 (m, 2H, -Ar-*p*-H). ^13^C{^1^H} NMR (101 MHz, CDCl_3_): δ = 16.9 (-S*C*H_3_); 55.8 (-O*C*H_3_); 111.7 (-Ar-*m*-C); 115.1 (=*C*H); 120.8 (-Ar-*m*-C); 127.9 (-Ar-C1); 131.4 (-Ar-*m*-C); 132.2 (-Ar-*o*-C); 157.2 (-Ar-*C*-OCH_3_); 178.1 (-*C*-O-); 181.4 (-*C*=S). MS (EI): *m*/*z* = 536 [M(^58^Ni)]^•+^. Elemental analysis: calculated for C_22_H_22_O_4_NiS_4_ C: 49.18 %; H: 4.13 %, found: C: 49.45 %; H: 4.15 %.

[Ni(1-(4-methoxyphenyl)-3-(methylthio)-3-thioxo-prop-1-en-1-olate-*O*,*S*)] (Ni3)

Synthesis was performed according to general procedure 1. NiCl_2_·6H_2_O (250 mg, 1 mmol) and L3 (506 mg, 2 mmol) were used. Yield: 350 mg (65.3 %) as red crystals. ^1^H NMR (400 MHz, CDCl_3_): δ = 2.63 (s, 6H, -SC*H*_3_); 3.89 (s, 6H, -OC*H*_3_); 6.92 (d, ^3^*J_H-H_*=8.8 Hz, 4H, -Ar-*o*-H); 7.10 (s, 2H, =C*H*); 7.88 (d, ^3^*J_H-H_*=8.9 Hz, 4H, -Ar-*m*-H). ^13^C{^1^H} NMR (101 MHz, CDCl_3_): δ = 16.6 (-S*C*H_3_); 55.5 (-O*C*H_3_); 110.0 (=*C*H); 113.8 (-Ar-*o*-C); 129.4 (-Ar-*m*-C); 130.3 (-Ar-C1); 162.5 (-Ar-*C*-OCH_3_); 177.8 (-*C*-O-); 181.6 (-*C*=S). MS (EI): *m*/*z* = 536 [M(^58^Ni)]^•+^. Elemental analysis: calculated for C_22_H_22_O_4_NiS_4_ C: 49.18 %; H: 4.13 %, found: C: 49.26 %; H: 4.10 %.

[Ni(1-(2-ethoxyphenyl)-3-(methylthio)-3-thioxo-prop-1-en-1-olate-*O*,*S*)] (Ni4)

Synthesis was performed according to general procedure 1. NiCl_2_·6H_2_O (250 mg, 1 mmol) and L4 (540 mg, 2 mmol) were used. Yield: 250 mg (44.3 %) as red crystals. ^1^H NMR (400 MHz, CDCl_3_): δ = 1.48 (m, 6H, -OCH_2_C*H*_3_); 2.61 (t, 6H, -CH_3_); 4.11 (q, ^3^*J_H-H_*=6.9 Hz, 4H, -OC*H*_2_CH_3_); 6.90-6.96 (m, 4H, -Ar-*m*-H); 7.28-7.46 (m, 2H, -Ar-*p*-H); 7.79-7.81 (m, 2H, -Ar-*o*-H). ^13^C{^1^H} NMR (101 MHz, CDCl_3_): δ = 14.9 (-OCH_2_*C*H_3_); 16.7 (-*C*H_3_); 64.5 (-O*C*H_2_CH_3_); 112.8 (-Ar-*m*-C); 115.0 (=*C*H); 120.7 (-Ar-*m*-C); 127.8 (=*C*-OH); 131.6 (-Ar-*o*-C); 132.3 (-Ar-*p*-C); 156.7 (qC, -Ar-*o*-C); 177.9 (Ar-C1); 181.1 (-*C*=S). MS (EI): *m*/*z* = 564 [M(^58^Ni)]^•+^. Elemental analysis: calculated for C_24_H_26_O_4_NiS_4_ C: 50.98 %; H: 4.64 %, found: C: 50.99 %; H: 4.63 %.

[Ni(1-(3-ethoxyphenyl)-3-(methylthio)-3-thioxo-prop-1-en-1-olate-*O*,*S*)] (Ni5)

Synthesis was performed according to general procedure 1. NiCl_2_·6H_2_O (250 mg, 1 mmol) and L5 (540 mg, 2 mmol) were used. Yield: 200 mg (35.5 %) as red crystals. ^1^H NMR (400 MHz, CDCl_3_): δ = 1.57 (m, 6H, -OCH_2_C*H*_3_); 2.66 (t, 6H, -CH_3_); 4.07 (q, ^3^*J_H-H_*=7.1 Hz, 4H, -OC*H*_2_CH_3_); 7.08 (d, 2H, ^3^*J_H-H_*=8.0 Hz, -Ar-*p*-H); 7.15 (s, 2H, =C*H*); 7.33 (m, 2H, -Ar-*m*-H); 7.41 (s, 2H, -Ar-*o*-H); 7.47 (d, 2H, ^3^*J_H-H_*=7.8 Hz, -Ar-*o*-H). ^13^C{^1^H} NMR (101 MHz, CDCl_3_): δ = 14.7 (-OCH_2_*C*H_3_); 17.1 (-*C*H_3_); 63.7 (-O*C*H_2_CH_3_); 108.0 (=*C*H); 112.4 (-Ar-*o*-C); 118.3 (-Ar-*p*-C); 118.9 (-Ar-*o*-C); 129.7 (-Ar-*m*-C); 159.2 (qC, -Ar-*m*-C); 169.1 (Ar-C1); 217.2 (-*C*=S). MS (EI): *m*/*z* = 564 [M(^58^Ni)]^•+^. Elemental analysis: calculated for C_24_H_26_O_4_NiS_4_ C: 50.98 %; H: 4.64 %, found: C: 51.08 %; H: 4.63 %.

[Ni(1-(4-ethoxyphenyl)-3-(methylthio)-3-thioxo-prop-1-en-1-olate-*O*,*S*)] (Ni6)

Synthesis was performed according to general procedure 1. NiCl_2_·6H_2_O (250 mg, 1 mmol) and L6 (540 mg, 2 mmol) were used. Yield: 190 mg (33.7 %) as red crystals. ^1^H NMR (400 MHz, CDCl_3_): δ = 1.37 (t, ^3^*J_H-H_*=6.9 Hz, 6H, -OCH_2_C*H*_3_); 2.53 (s, 6H, -CH_3_); 4.02 (q, ^3^*J_H-H_*=6.9 Hz, 4H, -OC*H*_2_CH_3_); 6.81 (d, 4H, ^3^*J_H-H_*=8.6 Hz, -Ar-*m*-H); 7.01 (s, 2H, =C*H*); 7.78 (d, 4H, ^3^*J_H-H_*=8.6 Hz, -Ar-*o*-H). ^13^C{^1^H} NMR (101 MHz, CDCl_3_): δ = 14.7 (-OCH_2_*C*H_3_); 16.6 (-*C*H_3_); 63.7 (-O*C*H_2_CH_3_); 110.0 (=*C*H); 114.2 (-Ar-*m*-C); 130.1 (-Ar-C1); 162.0 (-Ar-*C*-OCH_3_); 177.8 (-*C*-O-); 181.5 (-*C*=S). MS (EI): *m*/*z* = 564 [M(^58^Ni)]^•+^. Elemental analysis: calculated for C_24_H_26_O_4_NiS_4_ C: 50.98 %; H: 4.64 %, found: C: 50.91 %; H: 4.60 %.

[Pd(1-(2-methoxyphenyl)-3-(methylthio)-3-thioxo-prop-1-en-1-olate-*O*,*S*)] (Pd1)

Synthesis was performed according to general procedure 1. (PhCN)_2_PdCl_2_ (399 mg, 1 mmol) and L1 (500 mg, 2 mmol) were used. Yield: 120 mg (20.5 %) as red crystals. ^1^H NMR (400 MHz, CDCl_3_): δ = 2.66 (s, 6H, -SC*H*_3_); 3.90 (s, 6H, -OC*H*_3_); 6.93-7.01 (m, 4H, -Ar-*m*-H/-Ar-*o*-H); 7.19 (s, 2H, =C*H*); 7.42 (m, 2H, -Ar-*p*-H); 7.79 (d, ^3^*J_H-H_*=7.6 Hz, ^4^*J_H-H_*=1.8 Hz, 2H, -Ar-*m*-H). ^13^C{^1^H} NMR (101 MHz, CDCl_3_): δ = 17.5 (-S*C*H_3_); 55.9 (-O*C*H_3_); 111.8 (-Ar-*m*-C); 115.3 (=*C*H); 120.7 (-Ar-*m*-C); 129.0 (-Ar-C1); 130.4 (-Ar-*m*-C); 131.2 (-Ar-*o*-C); 157.4 (-Ar-*C*-OCH_3_); 179.2 (-*C*-O-); 180.2 (-*C*=S). MS (EI): *m*/*z* = 564 [M(^106^Pd)]^•+^. Elemental analysis: calculated for C_22_H_22_O_4_PdS_4_ C: 45.16 %; H: 3.79 %, found: C: 45.23 %; H: 3.72 %.

[Pd(1-(3-methoxyphenyl)-3-(methylthio)-3-thioxo-prop-1-en-1-olate-*O*,*S*)] (Pd2)

Synthesis was performed according to general procedure 1. (PhCN)_2_PdCl_2_ (399 mg, 1 mmol) and L2 (500 mg, 2 mmol) were used. Yield: 150 mg (25.7 %) as red crystals. ^1^H NMR (400 MHz, CDCl_3_): δ = 2.71 (s, 6H, -SC*H*_3_); 4.09 (s, 6H, -OC*H*_3_); 7.09 (dd, ^3^*J_H-H_*=8.1 Hz, ^4^*J_H-H_*=2.4 Hz, 2H, -Ar-*o*-H); 7.16 (s, 2H, =C*H*); 7.34 (m, 2H, -Ar-*m*-H); 7.54-7.57 (m, 2H, -Ar-*p*-H); 7.65 (m, 2H, -Ar-*o*-H). ^13^C{^1^H} NMR (101 MHz, CDCl_3_): δ = 17.5 (-S*C*H_3_); 55.4 (-O*C*H_3_); 110.8 (-Ar-*m*-C); 113.0 (=*C*H); 117.9 (-Ar-*m*-C); 120.0 (-Ar-C1); 129.4 (-Ar-*m*-C); 139.7 (-Ar-*o*-C); 159.8 (-Ar-*C*-OCH_3_); 178.1 (-*C*-O-). MS (EI): *m*/*z* = 564 [M(^106^Pd)]^•+^. Elemental analysis: calculated for C_22_H_22_O_4_PdS_4_ C: 45.16 %; H: 3.79 %, found: C: 45.40 %; H: 3.75 %.

[Pd(1-(4-methoxyphenyl)-3-(methylthio)-3-thioxo-prop-1-en-1-olate-*O*,*S*)] (Pd3)

Synthesis was performed according to general procedure 1. (PhCN)_2_PdCl_2_ (399 mg, 1 mmol) and L3 (500 mg, 2 mmol) were used. Yield: 60 mg (10.3 %) as red crystals. ^1^H NMR (400 MHz, CDCl_3_): δ = 2.70 (s, 6H, -SC*H*_3_); 3.91 (s, 6H, -OC*H*_3_); 6.97 (d, ^3^*J_H-H_*=8.7 Hz, 4H, -Ar-*o*-H); 7.13 (s, 2H, =C*H*); 8.02 (d, ^3^*J_H-H_*=9.1 Hz, 4H, -Ar-*m*-H). ^13^C{^1^H} NMR (101 MHz, CDCl_3_): δ = 17.4 (-S*C*H_3_); 55.5 (-O*C*H_3_); 110.5 (=*C*H); 113.9 (-Ar-*o*-C); 130.0 (-Ar-*m*-C); 130.3 (-Ar-C1); 162.7 (-Ar-*C*-OCH_3_); 178.1 (-*C*-O-); 180.4 (-*C*=S). MS (EI): *m*/*z* = 564 [M(^106^Pd)]^•+^. Elemental analysis: calculated for C_22_H_22_O_4_PdS_4_ C: 45.16 %; H: 3.79 %, found: C: 45.23 %; H: 3.80 %.

[Pd(1-(2-ethoxyphenyl)-3-(methylthio)-3-thioxo-prop-1-en-1-olate-*O*,*S*)] (Pd4)

Synthesis was performed according to general procedure 1. (PhCN)_2_PdCl_2_ (600 mg, 1.5 mmol) and L4 (700 mg, 2.9 mmol) were used. Yield: 80 mg (13.1 %) as red crystals. ^1^H NMR (400 MHz, CDCl_3_): δ = 1.28 (m, 6H, -OCH_2_C*H*_3_); 2.66 (s, 6H, -CH_3_); 4.17 (q, ^3^*J_H-H_*=7.0 Hz, 4H, -OC*H*_2_CH_3_); 6.95-7.02 (m, 4H, -Ar-*m*-H); 7.44-7.61 (m, 4H, -Ar-*p*-H/=C*H*); 7.75-7.84 (m, 2H, -Ar-*o*-H). ^13^C{^1^H} NMR (101 MHz, CDCl_3_): δ = 14.9 (-OCH_2_*C*H_3_); 16.7 (-*C*H_3_); 64.5 (-O*C*H_2_CH_3_); 112.8 (-Ar-*m*-C); 115.0 (=*C*H); 120.7 (-Ar-*m*-C); 127.8 (=*C*-OH); 131.6 (-Ar-*o*-C); 132.3 (-Ar-*p*-C); 156.7 (qC, -Ar-*o*-C); 177.9 (Ar-C1); 181.1 (-*C*=S). No EI mass spectrum could be obtained. Elemental analysis: calculated for C_24_H_26_O_4_PdS_4_ C: 47.02 %; H: 4.26 %, found: C: 47.41 %; H: 4.26 %.

[Pd(1-(3-ethoxyphenyl)-3-(methylthio)-3-thioxo-prop-1-en-1-olate-*O*,*S*)] (Pd5)

Synthesis was performed according to general procedure 1. (PhCN)_2_PdCl_2_ (399 mg, 1 mmol) and L5 (500 mg, 2.1 mmol) were used. Yield: 120 mg (16.6 %) as red crystals. ^1^H NMR (400 MHz, CDCl_3_): δ = 1.46 (t, ^3^*J_H-H_*=7.0 Hz, 6H, -OCH_2_C*H*_3_); 2.71 (s, 6H, -CH_3_); 4.13 (q, ^3^*J_H-H_*=7.0 Hz, 4H, -OC*H*_2_CH_3_); 7.07-7.09 (m, 2H, -Ar-*p*-H); 7.13 (s, 2H, =C*H*); 7.32 (m, 2H, -Ar-*m*-H); 7.54-7.58 (m, 4H, -Ar-*o*-H). ^13^C{^1^H} NMR (101 MHz, CDCl_3_): δ = 14.8 (-OCH_2_*C*H_3_); 17.5 (-*C*H_3_); 63.6 (-O*C*H_2_CH_3_); 110.9 (=*C*H); 113.5 (-Ar-*o*-C); 118.5 (-Ar-*p*-C); 120.0 (-Ar-*o*-C); 129.4 (-Ar-*m*-C); 159.1 (qC, -Ar-*m*-C); 178.4 (Ar-C1); 182.6 (-*C*=S). MS (EI): *m*/*z* = 612 [M(^106^Pd)]^•+^. Elemental analysis: calculated for C_24_H_26_O_4_PdS_4_ C: 47.02 %; H: 4.27 %, found: C: 46.68 %; H: 4.24 %.

[Pd(1-(4-ethoxyphenyl)-3-(methylthio)-3-thioxo-prop-1-en-1-olate-*O*,*S*)] (Pd6)

Synthesis was performed according to general procedure 1. (PhCN)_2_PdCl_2_ (399 mg, 1 mmol) and L6 (500 mg, 2.1 mmol) were used. Yield: 100 mg (16.3 %) as red crystals. ^1^H NMR (400 MHz, CDCl_3_): δ = 1.47 (t, ^3^*J_H-H_*=6.9 Hz, 6H, -OCH_2_C*H*_3_); 2.70 (s, 6H, -CH_3_); 4.14 (q, ^3^*J_H-H_*=6.9 Hz, 4H, -OC*H*_2_CH_3_); 6.95 (d, 4H, ^3^*J_H-H_*=8.8 Hz, -Ar-*m*-H); 7.13 (s, 2H, =C*H*); 8.00 (d, 4H, ^3^*J_H-H_*=8.9 Hz, -Ar-*o*-H). ^13^C{^1^H} NMR (101 MHz, CDCl_3_): δ = 14.7 (-OCH_2_*C*H_3_); 17.4 (-*C*H_3_); 63.7 (-O*C*H_2_CH_3_); 110.0 (=*C*H); 114.3 (-Ar-*m*-C); 130.0 (-Ar-C1). MS (EI): *m*/*z* = 612 [M(^106^Pd)]^•+^. Elemental analysis: calculated for C_24_H_26_O_4_PdS_4_ C: 47.02 %; H: 4.27 %, found: C: 47.35 %; H: 4.32 %.

[Pt(1-(2-methoxyphenyl)-3-(methylthio)-3-thioxo-prop-1-en-1-olate-*O*,*S*)] (Pt1)

Synthesis was performed according to general procedure 1. (PhCN)_2_PtCl_2_ (492 mg, 1 mmol) and L1 (500 mg, 1 mmol) were used. Yield: 60 mg (8.9 %) as red crystals. ^1^H NMR (400 MHz, CDCl_3_): δ = 2.63 (s, 6H, -SC*H*_3_); 3.89 (s, 6H, -OC*H*_3_); 6.92-7.11 (m, 4H, -Ar-*m*-H/-Ar-*o*-H); 7.20 (s, 2H, =C*H*); 7.43-7.54 (m, 2H, -Ar-*p*-H); 7.82 (dd, ^3^*J_H-H_*=7.6 Hz, ^4^*J_H-H_*=1.7 Hz, 2H, -Ar-*m*-H). ^13^C{^1^H} NMR (101 MHz, CDCl_3_): δ = 17.5 (-S*C*H_3_); 55.8 (-O*C*H_3_); 111.9 (-Ar-*m*-C); 117.1 (=*C*H); 120.4 (-Ar-*m*-C); 129.1 (-Ar-C1); 130.4 (-Ar-*m*-C); 131.9 (-Ar-*o*-C). MS (EI): *m*/*z* = 673 [M(^195^Pt)]^•+^. Elemental analysis: calculated for C_22_H_22_O_4_PtS_4_ C: 39.22 %; H: 3.29 %, found: C: 39.36 %; H: 3.32 %.

[Pt(1-(3-methoxyphenyl)-3-(methylthio)-3-thioxo-prop-1-en-1-olate-*O*,*S*)] (Pt2)

Synthesis was performed according to general procedure 1. (PhCN)_2_PtCl_2_ (470 mg, 1 mmol) and L2 (506 mg, 1 mmol) were used. Yield: 290 mg (43.1 %) as red crystals. ^1^H NMR (400 MHz, CDCl_3_): δ = 2.66 (s, 6H, -SC*H*_3_); 3.91 (s, 6H, -OC*H*_3_); 7.12 (dd, ^3^*J_H-H_*=8.4 Hz, ^4^*J_H-H_*=2.5 Hz, 2H, -Ar-*o*-H); 7.14 (s, 2H, =C*H*); 7.32 (m, 2H, -Ar-*m*-H); 7.59 (d, ^3^*J_H-H_*=7.8 Hz, 2H, -Ar-*p*-H); 7.67 (m, 2H, -Ar-*o*-H). ^13^C{^1^H} NMR (101 MHz, CDCl_3_): δ = 17.6 (-S*C*H_3_); 55.4 (-O*C*H_3_); 112.5 (-Ar-*m*-C); 112.5 (=*C*H); 117.5 (-Ar-*m*-C); 119.4 (-Ar-C1); 129.6 (-Ar-*m*-C); 140.3 (-Ar-*o*-C); 160.0 (-Ar-*C*-OCH_3_). MS (EI): *m*/*z* = 673 [M(^195^Pt)]^•+^. Elemental analysis: calculated for C_22_H_22_O_4_PtS_4_ C: 39.22 %; H: 3.29 %, found: C: 39.11 %; H: 3.25 %.

[Pt(1-(4-methoxyphenyl)-3-(methylthio)-3-thioxo-prop-1-en-1-olate-*O*,*S*)] (Pt3)

Synthesis was performed according to general procedure 1. (PhCN)_2_PtCl_2_ (470 mg, 1 mmol) and L3 (506 mg, 1 mmol) were used. Yield: 210 mg (31.2 %) as red crystals. ^1^H NMR (400 MHz, CDCl_3_): δ = 2.72 (s, 6H, -SC*H*_3_); 3.90 (s, 6H, -OC*H*_3_); 7.00 (d, ^3^*J_H-H_*=9.0 Hz, 4H, -Ar-*o*-H); 7.02 (s, 2H, =C*H*); 8.04 (d, ^3^*J_H-H_*=8.9 Hz, 4H, -Ar-*m*-H). ^13^C{^1^H} NMR (101 MHz, CDCl_3_): δ = 17.5 (-S*C*H_3_); 55.5 (-O*C*H_3_); 112.0 (=*C*H); 114.1 (-Ar-*o*-C); 129.3 (-Ar-*m*-C); 131.4 (-Ar-C1); 162.3 (-Ar-*C*-OCH_3_); 173.6 (-*C*-O-). MS (EI): *m*/*z* = 673 [M(^195^Pt)]^•+^. Elemental analysis: calculated for C_22_H_22_O_4_PtS_4_ C: 39.22 %; H: 3.29 %, found: C: 39.29 %; H: 3.32 %.

[Pt(1-(2-ethoxyphenyl)-3-(methylthio)-3-thioxo-prop-1-en-1-olate-*O*,*S*)] (Pt4)

Synthesis was performed according to general procedure 1. (PhCN)_2_PtCl_2_ (700 mg, 1.5 mmol) and L4 (700 mg, 2.9 mmol) were used. Yield: 120 mg (17.1 %) as red crystals. ^1^H NMR (400 MHz, CDCl_3_): δ = 1.28 (m, ^3^*J_H-H_*=7.2 Hz, 6H, -OCH_2_C*H*_3_); 2.66 (s, 6H, -CH_3_); 4.17 (q, ^3^*J_H-H_*=7.2 Hz, 4H, -OC*H*_2_CH_3_); 6.95-7.02 (m, 4H, -Ar-*m*-H); 7.44-7.50 (m, 4H, -Ar-*p*-H/=C*H*); 7.75-7.83 (m, 2H, -Ar-*o*-H). ^13^C{^1^H} NMR (101 MHz, CDCl_3_): δ = 14.9 (-OCH_2_*C*H_3_); 16.7 (-*C*H_3_); 64.5 (-O*C*H_2_CH_3_); 112.3 (-Ar-*m*-C); 115.0 (=*C*H); 120.7 (-Ar-*m*-C); 127.8 (=*C*-OH); 131.6 (-Ar-*o*-C); 132.3 (-Ar-*p*-C); 156.7 (qC, -Ar-*o*-C); 177.9 (Ar-C1); 181.1 (-*C*=S). No EI mass spectrum could be obtained. Elemental analysis: calculated for C_24_H_26_O_4_PtS_4_ C: 41.08 %; H: 3.73 %, found: C: 41.18 %; H: 3.25 %.

[Pt(1-(3-ethoxyphenyl)-3-(methylthio)-3-thioxo-prop-1-en-1-olate-*O*,*S*)] (Pt5)

Synthesis was performed according to general procedure 1. (PhCN)_2_PtCl_2_ (470 mg, 1 mmol) and L5 (510 mg, 2 mmol) were used. Yield: 210 mg (30.0 %) as red crystals. ^1^H NMR (400 MHz, CDCl_3_): δ = 1.45 (t, ^3^*J_H-H_*=7.1 Hz, 6H, -OCH_2_C*H*_3_); 2.65 (s, 6H, -CH_3_); 4.13 (q, ^3^*J_H-H_*=6.9 Hz, 4H, -OC*H*_2_CH_3_); 7.09-7.13 (m, 4H, -Ar-*p*-H/ =C*H*); 7.31 (m, 2H, -Ar-*m*-H); 7.58-7.60 (m, 4H, -Ar-*o*-H). ^13^C{^1^H} NMR (101 MHz, CDCl_3_): δ = 14.8 (-OCH_2_*C*H_3_); 17.6 (-*C*H_3_); 63.6 (-O*C*H_2_CH_3_); 112.5 (=*C*H); 112.9 (-Ar-*o*-C); 118.0 (-Ar-*p*-C); 119.4 (-Ar-*o*-C); 129.5 (-Ar-*m*-C); 159.3 (qC, -Ar-*m*-C); 177.8 (Ar-C1). MS (EI): *m*/*z* = 701 [M(^195^Pt)]^•+^. Elemental analysis: calculated for C_24_H_26_O_4_PtS_4_ C: 41.08 %; H: 3.73 %, found: C: 41.03 %; H: 3.78 %.

### 3.3. Crystal Structure Determination

A Nonius KappaCCD diffractometer and graphite-monochromated Mo-K_α_ radiation were used to collect intensity data for the compounds. Corrections were performed for polarization and Lorentz effects, and absorption was taken into account on a semi-empirical basis using multiple scans [71,72]. Direct methods (SHELXS) were used to solve structures, which were refined by full-matrix least squares techniques against Fo^2^ (SHELXL-97) [73]. The hydrogen atoms of Ni6 were located by difference Fourier synthesis and refined isotropically. All other hydrogen atoms were included at calculated positions with fixed thermal parameters. Crystallographic data, structure solution, and refinement details are summarized in Appendix A. Structure representations were produced with MERCURY [74]. Supporting information available: crystallographic data (excluding structure factors) have been deposited with the Cambridge Crystallographic Data Centre as supplementary publication CCDC-1953242 for Ni1, CCDC-1953243 for Ni3, CCDC-1953244 for Ni4, CCDC-1953245 for Ni6, and CCDC-1953246 for Pd1, and CCDC-1953247 for Ptdmso8. Copies of the data can be obtained free of charge on application to CCDC, 12 Union Road, Cambridge CB2 1EZ, UK [E-mail: deposit@ccdc.cam.ac.uk].

### 3.4. Stability Determinations

NMR spectra were measured on a Bruker Avance 400 MHz system. Substances were solved in DMSO-d_6_ or CD_2_Cl_2_ and measured directly at 37 °C or room temperature for 72 h. NS = 128 scans, t = 709 s/2891 seconds break, 72 measurements.

For UV–VIS spectroscopy, a JASCO UV–VIS V-760 spectrometer was used. Spectra were measured between 240 and 800 nm at 1 nm steps with a scan speed of 400 nm/min. Compound measurements at 100 µM concentration were normalized to the respective buffer.

### 3.5. Biological Assays

Cell cultures were kept under standard conditions (5 % CO_2_, 37 °C, 90% humidity) in an RPMI medium with 10% FCS, 100 µg/mL streptomycin, and 100 U/mL penicillin (Life Technologies, Darmstadt, Germany). Reference cisplatin (Sigma, Taufkirchen, Germany) was dissolved freshly in 0.9% NaCl solution at a concentration of 1 mg/mL, and diluted appropriately. Described metal(II) complexes and their ligands were dissolved in DMSO. Platinum-resistant A2780 and SKOV3 cells were established as described [18]. IC_50_ values were determined using the CellTiter96 non-radioactive proliferation assay (MTT assay, Promega, Walldorf, Germany). A total of 5000 cells were allowed to attach per well of 96-well plates for 24 h and treated for 48 h with different concentrations of the substances (ligands tests: 0, 1, 10, 50, 100, 500, 1000 µm) and for cisplatin and metal complexes from 0 to 100 µM (0.1, 1, 5, 10, 50, 100 µM). Each measurement was performed in triplicate and repeated three times. The amount of metabolic active cells was quantified using the MTT assay. Relative values compared to the mean of medium controls were calculated after background subtraction. Non-linear regression analyses were conducted in GraphPad 5.0 software using the Hill slope.

## 4. Conclusions

Overall, we report on 17 novel metal complexes with *O*,*S* ligands, and include a comparison with previously reported results on other platinum(II) molecules, as well as ruthenium(II) and osmium(II) counterparts [18,49,50]. The bidendate compounds were characterized using classical methods, including NMR spectroscopy, MS spectrometry, elemental analysis, and some molecular structures. Stability determinations show stable compounds in DMSO for palladium(II) and nickel(II) complexes; molecular structures show a cis-geometry for all square-planar measured metal(II) complexes. The comparison of NMR spectra and molecular structures shows both characteristic changes after complexation of the β-hydroxydithiocinnamic acid esters to the metal(II) center, resulting in an elongation of the -C-S bonds of the thiocarbonyl groups, and a shortening of the -C-O-bonds. SAR analyses regarding the metal ion (M), the alkyl-chain position (P), and the length (L) revealed the following order of effect strength for in vitro activity: M > P > L. In general, the highest activities have Pd complexes and ortho-substituted compounds. The analysis of IC_50_ values shows promising results for the palladium(II) complexes, as some of them show lower values compared to cisplatin and are able to elude cisplatin resistance mechanisms in ovarian cancer cell lines. Therefore, the most active compound Pd3 will be further investigated in vivo. 

## Data Availability

The data presented in this study are available in the article and in the Appendix A. Crystallographic data (excluding structure factors) have been deposited with the Cambridge Crystallographic Data Centre as supplementary publication CCDC-1953242 for Ni1, CCDC-1953243 for Ni3, CCDC-1953244 for Ni4, CCDC-1953245 for Ni6, CCDC- 1953246 for Pd1, and CCDC-1953247 for Ptdmso8. Copies of the data can be obtained free of charge on application to CCDC, 12 Union Road, Cambridge CB2 1EZ, UK [E-mail: deposit@ccdc.cam.ac.uk].

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
