# Peer review of "Novel Nickel(II), Palladium(II), and Platinum(II) Complexes with O,S Bidendate Cinnamic Acid Ester Derivatives: An In Vitro Cytotoxic Comparison to Ruthenium(II) and Osmium(II) Analogues"

_ijms, 2022, doi:10.3390/ijms23126669_

Round 1

Reviewer 1 Report

This manuscript by Jana Hildebrandt et al. is well-written and based on sound scientific evidence. The results are not particularly impressive, but certainly deserve publication.

The only major reservation is in the .CIF files where several A and B level alerts are present. The authors provided standard replies in all cases of concern. Unfortunately, I’m not an expert on crystallography. In my opinion the .CIF files should be checked and verified by people with suitable experience. I was also unable to find in the provided SI additional figures, such as Figure S3.

All complexes are depicted as square-planar or distorted square-planar. In all cases isomers with cis arrangement of the sulfur and oxygen donor atoms are present. Do you any explanation why the other plausible square-planar isomer is not formed? Do you see any other products in the reaction mixtures? The yields are often low to moderate. It would be interesting if you commented on this issue in the revised manuscript.

Some minor problems:

Please explain explicitly “resistance factors”

Line 332:  “All reactions were performed using standard conditions.” This sentence is meaningless, please give some details or remove.

Please explain abbreviations “MS (DEI)”

Line 533: “NMR spectra were measured via NMR spectroscopy” What do you mean in this sentence?

Author Response

This manuscript by Jana Hildebrandt et al. is well-written and based on sound scientific evidence. The results are not particularly impressive, but certainly deserve publication.

The only major reservation is in the .CIF files where several A and B level alerts are present. The authors provided standard replies in all cases of concern. Unfortunately, I’m not an expert on crystallography. In my opinion the .CIF files should be checked and verified by people with suitable experience. I was also unable to find in the provided SI additional figures, such as Figure S3.

All complexes are depicted as square-planar or distorted square-planar. In all cases isomers with cis arrangement of the sulfur and oxygen donor atoms are present. Do you any explanation why the other plausible square-planar isomer is not formed? Do you see any other products in the reaction mixtures? The yields are often low to moderate. It would be interesting if you commented on this issue in the revised manuscript.

Some minor problems:

Please explain explicitly “resistance factors”

Authors reply: We have added the equitation for the resistance factor on page 6 (RF = IC50resistant cells / IC50sensitive cells)

Line 332:  “All reactions were performed using standard conditions.” This sentence is meaningless, please give some details or remove.

Authors reply: We have removed this sentence.

Please explain abbreviations “MS (DEI)”

Authors reply: We added the explanation at the first occurrence MS (DEI; Desorption Electron Impact Mass Spectrometry)

Line 533: “NMR spectra were measured via NMR spectroscopy” What do you mean in this sentence?

Authors reply: We changed this sentence to “NMR spectra were measured on a Bruker Avance 400 MHz system”

Reviewer 2 Report

The manuscript by Hildebrandt et al. presents a large family of mononuclear complexes of nickel, palladium and platinum, derived from O,S-bidentate cinnamic ester derivatives, with very promising results in the search for effective chemotherapy for forms of ovarian cancers in desperate need for new therapeutics.

The preparation and characterization of the complexes is presented herein, together with a survey of their effects on various cancer and healthy cell lines, with an emphasis on a search of cytotoxic activity on cells having developed some form of resistance to cis-platin. This is a very valuable quest.

From a format perspective, the manuscript is well-written, although it would benefit from another round of proof-reading by native English speakers (mostly to help identify when commas, or articles, are required or not, and advise on the choice of words and past tenses). The authors are encouraged to look into this before submitting their revised manuscript.

A few other format clarifications would be useful. Page 3, line 112: ‘220,000 new cases annually’. Is this world-wide? If not, which country/continent? Page 4, Table 1, when reporting chemical shifts, please indicate solvent, temperature and magnetic field. Page 8, Fig 3A, the colours for the legends are quite similar, hence they are difficult to differentiate. Page 10, line 332, what are ‘standard conditions’ for reactions?

 From a content perspective, a few points deserve some consideration, in order to best benefit an interested reader.

1) In the background section, page 2, line 49, it is stated that carboplatin and oxaliplatin are affected by the same resistance mechanisms. Could the authors expand on this claim? Although cisplatin and carboplatin indeed lead to the exact same molecular adducts with DNA, and therefore similar resistance is anticipated, would oxaliplatin not ‘leave behind’ a much more hydrophobic adduct onto the DNA helix, and therefore possibly generate different responses, including resistance mechanisms? It may be useful to provide a little bit more context to the reader. See for instance https://doi.org/10.1016/j.ctrv.2007.01.009

The authors are encouraged to clarify their original statement for the reader’s benefit.

2) Although the choice of the ligand platform (beta-hydroxydithiocinnamic acid esters) seems to be mostly historical (it seems to be a strong expertise from that group), it would be very valuable if the authors provided some rationale for the use of these square planar complexes as anticancer drugs. Since they do not seem to have labile leaving groups, as in the cisPt/oxaliPt family, what do the authors imagine could be reasonable anticipated targets, and potential modes of action (or is it a ‘shot in the dark’?)? Obviously, precedents with the octahedral, mostly inert, osmium and ruthenium complexes are not transposable to these square planar complexes, with a large range of ligand exchange kinetics. Some insight into the possible anticipated mechanisms of cancer cell interference would be very valuable.

3) Coordination complexes are central to this article, hence more information around their structures would be highly valuable.

3a) In Figure 1 (and other figures throughout the article), the geometry that is represented is that of a ‘CIS’, or ‘Head-to-Head’ arrangement. In addition to what is observed in the crystallographic data reported herein, can the authors provide further experimental evidence, as well as theoretical explanation, behind such geometrical selection? Indeed, solid-state structures may be influenced by packing effects (and possibly the choice of the actual single crystal that was analyzed). Are there solution data that unambiguously rule out the coexistence of ‘TRANS’ or ‘Head-to-Tail’ arrangements? Would this be it solvent dependent?

3b) Page 24, lines 147 or so, the authors indicate that MS data show the molecular peak for each complex. Would it be possible to clarify this statement? The complexes of Ni(II), Pd(II) and Pt(II) with two chelating mono-anionic ligand should be neutral; how do they appear on the mass spectra, which report on charged compounds? Are these M-H+ species? Please clarify. Also, the experimental MS data reports several peaks for several compounds (including at lower m/z). Can the authors provide an assignment for the peak(s) of lower m/z values (is it a form of decomposition?)? In some cases, as for Pd4 (page 12, line 440), only a ‘mass’ of 164 is reported (which is unlikely the MH+); can the authors provide an explanation for these signals?

3c) The authors have paid attention to the important question of stability of these complexes, which is particularly critical with labile metal ions such as nickel(II), and semi-labile ions such as Pd(II). In this context, several points are worth clarifying.

First, DMSO-d6 is a good step forward in the study of complex stability, as it is a decent ligand competitor. Yet, in the context of cellular studies, aqueous buffers are even more important to consider. While not attempting to address the complexity of the RPMI medium, which contains electrolytes, aminoacids, sugars and other elements, it is important to assess how stable these complexes are in the presence of water, and most importantly high levels of salts such as chlorides (NaCl, KCl), as well as aminoacids such as hard one (aspartic acid, which may have some strong affinity for nickel(II)) and soft ones such as cysteine (ay have strong affinity to platinum). Since Fig S2 already indicates that the platinum complex, which one assumes should be the most kinetically inert, is sensitive to a competing ligand such as DMSO, these extended stability studies are essential to support the SAR analysis that the authors aim for in the cell studies. Information about the chemical stability of representative complexes to levels commensurate to what is found in RPMI would therefore be of great value (likely using a small percentage of DMSO for solubility purposes).

Second, Fig S2 presents interesting preliminary elements in the analysis of the chemical stability of the reported complexes. Starting from the bottom Pt complex, can the authors confirm the chemical structure of the ligand? Apparently, it is not L6, which should not display a triplet in the aromatic region, and should display Et chains in the aliphatic region (there is none in FigS2c). Furthermore, time t=0 in red already presents impurities. What are they? Are they observed in CDCl3? If that’s the compound that was submitted to cell studies, then the reader should be made aware that it is not pure, and the authors cautious about the associated conclusions. Finally on that complex, can the authors provide more information about the mechanisms of decomposition in DMSO? Have they monitored until all decomposition has been completed (stable NMR signals) and analyzed that final product? Again, it would be very valuable in order to assess if this is Pt or ligand specific. What happens with other ligands? If all the Pt complexes degrade similarly to the one displayed on Fig S2, the reader should be cautioned about the corresponding cell studies.

Thirdly about Fig. S2, can the authors explain the 1H NMR dynamics of the nickel complex (Fig S2a)? Indeed, out of the four aromatic protons, only three seem to have relatively resolved signals, and the 4th signal possibly corresponding to the broad signal around 8 ppm. In addition, the singlet is not clearly resolved, while a peak around 5.1 ppm is present. Can the authors provide some explanation behind the identity of these broad signals, and what they result from? In particular, can the authors rule out ligand dissociation, even at t=0 (nickel(II) being quite labile)? Again, this is critical for the understanding of the cell studies.

4) Cell studies. However tempting it can be in order to distill SAR trends, the value of calculating mean IC50 over a range of cell types remains questionable. This is mostly because cancer cells are so vastly different that one expects variations across cell responses. Calculating mean values then artificially creates very large standard deviations (SD) which then seem to invalidate the value of these compounds. For instance, looking at Figure 4A, assuming one can analyze a response as Mean +/- SD, the Pt derivatives would generate a mean response at around 19, and the SD bar goes to about 38. So that would mean Pt: 19 +/-19. Similarly, Ni: 8 +/-7, Pd: 5 +/3. These are not truly statistically different.

It seems that one is better off using the source data in Table 4 to anchor one’s discussion. The authors did a very good job highlighting promising behaviour using colour coding; it seems more judicious continuing along these lines (and discussing per cell lines), rather than calculating means (but it may just be a matter of preference).

Overall, this is a very interesting family of compounds. If the authors could provide (i) more support for the experimental isomers in solution, (ii) their stability in competing media (full characterization in DMSO, but also in the presence of high levels of chlorides, as well as in the presence of coordinating aminoacids), (iii) a rationale for anticipated cell targets, the manuscript would be even stronger.

Author Response

The manuscript by Hildebrandt et al. presents a large family of mononuclear complexes of nickel, palladium and platinum, derived from O,S-bidentate cinnamic ester derivatives, with very promising results in the search for effective chemotherapy for forms of ovarian cancers in desperate need for new therapeutics.

The preparation and characterization of the complexes is presented herein, together with a survey of their effects on various cancer and healthy cell lines, with an emphasis on a search of cytotoxic activity on cells having developed some form of resistance to cis-platin. This is a very valuable quest.

From a format perspective, the manuscript is well-written, although it would benefit from another round of proof-reading by native English speakers (mostly to help identify when commas, or articles, are required or not, and advise on the choice of words and past tenses). The authors are encouraged to look into this before submitting their revised manuscript.

Authors reply: We have carefully checked the manuscript for errors and spelling mistakes.

A few other format clarifications would be useful. Page 3, line 112: ‘220,000 new cases annually’. Is this world-wide? If not, which country/continent? Page 4, Table 1, when reporting chemical shifts, please indicate solvent, temperature and magnetic field. Page 8, Fig 3A, the colours for the legends are quite similar, hence they are difficult to differentiate. Page 10, line 332, what are ‘standard conditions’ for reactions?

Authors reply:  We added the information about the worldwide incidence and we removed the unspecific statement on page 10 about “standard conditions”

 From a content perspective, a few points deserve some consideration, in order to best benefit an interested reader.

1) In the background section, page 2, line 49, it is stated that carboplatin and oxaliplatin are affected by the same resistance mechanisms. Could the authors expand on this claim? Although cisplatin and carboplatin indeed lead to the exact same molecular adducts with DNA, and therefore similar resistance is anticipated, would oxaliplatin not ‘leave behind’ a much more hydrophobic adduct onto the DNA helix, and therefore possibly generate different responses, including resistance mechanisms? It may be useful to provide a little bit more context to the reader. See for instance https://doi.org/10.1016/j.ctrv.2007.01.009

The authors are encouraged to clarify their original statement for the reader’s benefit.

Authors reply: The reviewer points to an oversimplified statement in our manuscript. Indeed Oxaliplatin shows some differences compared to Cisplatin or Carboplatin. Mainly, adducts caused by Oxaliplatin have a different structure and are more bulky as other platinum adducts. Thus they are differentially recognized by DNA damage sensing proteins. However, clinical studies show that Oxaliplatin does not show a high activity on Cisplatin-resistant tumors. We changed our manuscript to reflect the knowledge more correctly and added following part (page 2): “Thus they are likely affected by the same resistance mechanisms and cause similar side effects, although Oxaliplatin may evade some resistance mechanisms [2, 7, 8]. Specifically Oxaliplatin adducts are more bulky, differentially recognized by DNA repair systems and not affected by mismatch repair deficiency caused resistance [8, 9]. However, a systematic review identified a poor response rate to Oxaliplatin in Cisplatin-resistant or – refractory cancers in accordance to most preclinical studies although some models highly resistant to Cisplatin show response to Oxaliplatin [10].”

2) Although the choice of the ligand platform (beta-hydroxydithiocinnamic acid esters) seems to be mostly historical (it seems to be a strong expertise from that group), it would be very valuable if the authors provided some rationale for the use of these square planar complexes as anticancer drugs. Since they do not seem to have labile leaving groups, as in the cisPt/oxaliPt family, what do the authors imagine could be reasonable anticipated targets, and potential modes of action (or is it a ‘shot in the dark’?)? Obviously, precedents with the octahedral, mostly inert, osmium and ruthenium complexes are not transposable to these square planar complexes, with a large range of ligand exchange kinetics. Some insight into the possible anticipated mechanisms of cancer cell interference would be very valuable.

Authors reply: Due to the square planar structure of the metal complexes, we assume an intercalation into the DNA, which has so far been assumed purely speculatively, which is why we have not dealt with it in more detail in the text. Further investigations into possible intercalations are to be carried out in the near future.

3) Coordination complexes are central to this article, hence more information around their structures would be highly valuable.

3a) In Figure 1 (and other figures throughout the article), the geometry that is represented is that of a ‘CIS’, or ‘Head-to-Head’ arrangement. In addition to what is observed in the crystallographic data reported herein, can the authors provide further experimental evidence, as well as theoretical explanation, behind such geometrical selection? Indeed, solid-state structures may be influenced by packing effects (and possibly the choice of the actual single crystal that was analyzed). Are there solution data that unambiguously rule out the coexistence of ‘TRANS’ or ‘Head-to-Tail’ arrangements? Would this be it solvent dependent?

Authors reply: That cis coordination mode formed exclusively we have observed since we were dealing nearly 30 years ago with these O,S ligands. A possible explanation is that the cis configurations are due to weak nonbonding sulfur-sulfur interactions leading to S...S distances between 300 to 310 pm that are significantly shorter than the sum of the van der Waals radii (ref 40:  lnorg. Chim. Acta 1998, 269, 83-90). Moreover, according to the “antisymbiosis” concept, two soft ligands (here: S donor atoms) in mutual trans position will have a destabilizing effect on each other. That phenomenon may be explained in terms of the different trans influences of O and S donors, respectively.

3b) Page 24, lines 147 or so, the authors indicate that MS data show the molecular peak for each complex. Would it be possible to clarify this statement? The complexes of Ni(II), Pd(II) and Pt(II) with two chelating mono-anionic ligand should be neutral; how do they appear on the mass spectra, which report on charged compounds? Are these M-H+ species? Please clarify. Also, the experimental MS data reports several peaks for several compounds (including at lower m/z). Can the authors provide an assignment for the peak(s) of lower m/z values (is it a form of decomposition?)? In some cases, as for Pd4 (page 12, line 440), only a ‘mass’ of 164 is reported (which is unlikely the MH+); can the authors provide an explanation for these signals?

Authors reply: We thank the reviewer for his detailed view onto the experimental data of the complexes. We mentioned in the text line 154 that next to the molecular ions (observed as radical cation during electron ionization EI) some fragments can be observed which clearly can be assigned to fragments from the ligands as figured out in the previous publication. For each complex we re-analyzed the spectra and compared each spectrum with a calculated isotope pattern. Luckily Ni, Pd and Pt as well show characteristic isotope patterns which irrevocable support the presence of the complexes. For a clear data presence, we mentioned only the principal ion, (most abundant ion) including the isotope number of the metal ion and termed it correctly as radical cation which corresponds to the primary product during electron ionization.

3c) The authors have paid attention to the important question of stability of these complexes, which is particularly critical with labile metal ions such as nickel(II), and semi-labile ions such as Pd(II). In this context, several points are worth clarifying.

First, DMSO-d6 is a good step forward in the study of complex stability, as it is a decent ligand competitor. Yet, in the context of cellular studies, aqueous buffers are even more important to consider. While not attempting to address the complexity of the RPMI medium, which contains electrolytes, aminoacids, sugars and other elements, it is important to assess how stable these complexes are in the presence of water, and most importantly high levels of salts such as chlorides (NaCl, KCl), as well as aminoacids such as hard one (aspartic acid, which may have some strong affinity for nickel(II)) and soft ones such as cysteine (ay have strong affinity to platinum). Since Fig S2 already indicates that the platinum complex, which one assumes should be the most kinetically inert, is sensitive to a competing ligand such as DMSO, these extended stability studies are essential to support the SAR analysis that the authors aim for in the cell studies. Information about the chemical stability of representative complexes to levels commensurate to what is found in RPMI would therefore be of great value (likely using a small percentage of DMSO for solubility purposes).

Authors reply: Thank you for that important comment and suggestions for further experiments. We are planning to carry out these experiments soon together with additional 3D cell culture or mouse models.

Second, Fig S2 presents interesting preliminary elements in the analysis of the chemical stability of the reported complexes. Starting from the bottom Pt complex, can the authors confirm the chemical structure of the ligand? Apparently, it is not L6, which should not display a triplet in the aromatic region, and should display Et chains in the aliphatic region (there is none in FigS2c). Furthermore, time t=0 in red already presents impurities. What are they? Are they observed in CDCl3? If that’s the compound that was submitted to cell studies, then the reader should be made aware that it is not pure, and the authors cautious about the associated conclusions. Finally on that complex, can the authors provide more information about the mechanisms of decomposition in DMSO? Have they monitored until all decomposition has been completed (stable NMR signals) and analyzed that final product? Again, it would be very valuable in order to assess if this is Pt or ligand specific. What happens with other ligands? If all the Pt complexes degrade similarly to the one displayed on Fig S2, the reader should be cautioned about the corresponding cell studies.

Authors reply: We are very grateful to the reviewer to draw attention to that error regarding the 1H NMR spectrum for Pt complex L6 in Figure S2c. Consequently, we have deleted this NMR spectrum from Figure S2.

Thirdly about Fig. S2, can the authors explain the 1H NMR dynamics of the nickel complex (Fig S2a)? Indeed, out of the four aromatic protons, only three seem to have relatively resolved signals, and the 4th signal possibly corresponding to the broad signal around 8 ppm. In addition, the singlet is not clearly resolved, while a peak around 5.1 ppm is present. Can the authors provide some explanation behind the identity of these broad signals, and what they result from? In particular, can the authors rule out ligand dissociation, even at t=0 (nickel(II) being quite labile)? Again, this is critical for the understanding of the cell studies.

Authors reply: A possible explanation could be that in the case of the Ni(II) complex and in contrast to the heavy homologues Pd(II) and Pt(II), an isomerization between square-planar and tetrahedral configuration occurs. The tetrahedral complex would then be paramagnetic, which explains the broad signals.

4) Cell studies. However tempting it can be in order to distill SAR trends, the value of calculating mean IC50 over a range of cell types remains questionable. This is mostly because cancer cells are so vastly different that one expects variations across cell responses. Calculating mean values then artificially creates very large standard deviations (SD) which then seem to invalidate the value of these compounds. For instance, looking at Figure 4A, assuming one can analyze a response as Mean +/- SD, the Pt derivatives would generate a mean response at around 19, and the SD bar goes to about 38. So that would mean Pt: 19 +/-19. Similarly, Ni: 8 +/-7, Pd: 5 +/3. These are not truly statistically different.

It seems that one is better off using the source data in Table 4 to anchor one’s discussion. The authors did a very good job highlighting promising behaviour using colour coding; it seems more judicious continuing along these lines (and discussing per cell lines), rather than calculating means (but it may just be a matter of preference).

Autors reply: The reviewer is correct that mean values are not ideal because of the large heterogeneity between cell lines. However, we think that in addition to the single values (Tab. 4) an overview of mean values (even if the comparison shows no statistically significant differences) is informative, because in the absence of any predictive marker for such metal-based compounds it reflects the clinical situation. We addressed this point in the revised version at page 6 (“In addition to the single IC50 values (Table 4) we calculated compound-specific or metal-specific mean values for all or specific groups of cell lines (Fig. 3, Fig. 4). The high variance of mean values caused by the heterogeneity of cell line specific sensitivity prevents significant differences. However, this may reflect the clinical situation if no predictive biomarker is available and the comparison of metal-specific mean values informs about the general effect of different metal ions.”)

Overall, this is a very interesting family of compounds. If the authors could provide (i) more support for the experimental isomers in solution, (ii) their stability in competing media (full characterization in DMSO, but also in the presence of high levels of chlorides, as well as in the presence of coordinating aminoacids), (iii) a rationale for anticipated cell targets, the manuscript would be even stronger.